# Influence of the Sputtering Temperature on Reflectivity and Electrical Properties of ITO/AgIn/ITO Composite Films for High-Reflectivity Anodes

**DOI:** 10.3390/ma16072849

**Published:** 2023-04-03

**Authors:** Xianqi Wang, Hongda Zhao, Bo Yang, Song Li, Zongbin Li, Haile Yan, Yudong Zhang, Claude Esling, Xiang Zhao, Liang Zuo

**Affiliations:** 1Key Laboratory for Anisotropy and Texture of Materials, School of Material Science and Engineering, Northeastern University, Shenyang 110819, China; 2Shenyang Dong Chuang Precious Metals Material Co., Ltd., Shenyang 110065, China; 3Laboratoire d’Étude des Microstructures et de Mécanique des Matériaux (LEM3), CNRS UMR 7239, Université de Lorraine, 57045 Metz, France

**Keywords:** sputtering temperature, ITO/AgIn/ITO composite films, optical properties, electrical properties, AgIn alloy films, magnetron sputtering

## Abstract

In this paper, indium tin oxide/silver indium/indium tin oxide (ITO/AgIn/ITO) composite films were deposited on glass substrates by magnetron sputtering. The effects of the sputtering temperature on the optical and electrical properties of the composite films were systematically investigated. The ITO/AgIn/ITO composite films deposited at sputtering temperatures of 25 °C and 100 °C demonstrated a high reflectivity of 95.3% at 550 nm and a resistivity of about 6.8–7.3 μΩ·cm. As the sputtering temperature increased, the reflectivity decreased and the resistivity increased slightly. The close connection between microstructure and surface morphology and the optical and electrical properties of the composite films was further illustrated by scanning electron microscopy imaging and atomic force microscopy imaging. It is shown that the ITO/AgIn/ITO thin films have a promising application for high-reflectivity anodes.

## 1. Introduction

High-reflectivity anodes are widely used in various optoelectronic applications including organic light-emitting devices (OLEDs) and photovoltaics [1,2,3,4]. The reflectivity of the bottom anode plays an essential role in the brightness of the top-emitting OLEDs. To date, plenty of materials, such as Al, Cu and Ag, have been proposed for the bottom anode. Among these materials, Ag exhibits the highest reflectivity in the visible wavelength range and the lowest resistivity. Nevertheless, Ag films are subject to agglomeration behavior when sputtered at higher temperatures [5,6].

In order to fundamentally solve the silver atom migration agglomeration problem, and simultaneously retain the performance advantages of silver thin films with high reflectivity and low resistivity, various alloying elements such as Mg, Pd, Cu, Al and other elements have been incorporated into the Ag layer [7,8,9,10]. Indium is a promising candidate as a doping material in the Ag layer. Jung et al. investigated the suppression of silver agglomeration behavior by doping of In atoms in silver–indium contacts. It was found that the silver contacts doped with In atoms demonstrated low resistivity (~3.8 × 10^−5^ Ω·cm^2^) and high reflectivity (~88.4% at 460 nm) after annealing [11]. Lee et al. studied the ITO ohmic contact layer and the AgIn reflector, which exhibited a low specific contact resistance (~1.90 × 10^−5^ Ω·cm^−2^) and a high visible light reflectance of ~84% [12]. In addition, indium has a large solid solution in silver, which can ensure the uniform sputtering of the target [13]. The silver–indium solid solution has good anti-tarnishing and mechanical properties, which can prevent warping of the film during large-area sputtering [14,15].

On the other hand, it is reported that covering the Ag layer with an ITO film can improve the hole injection to fulfill the performance requirements of high-reflectivity anodes [3,16]. ITO/Ag/ITO composite films have attracted much attention for their excellent optical and electrical properties, and have great potential for various optoelectronic applications [17,18,19,20]. Considerable efforts have also been made in the preparation of ITO/Ag/ITO thin films to achieve higher application requirements. However, most of them only study the effects of thickness, power, annealing, etc. on ITO/Ag/ITO thin films [21,22,23]. The effect of temperature on composite films has not been deeply studied. The temperature has a positive influence on the optoelectronic properties of ITO films [24,25]. In terms of the previous studies on ITO/Ag/ITO composite films, it is of special importance to study the effects of sputtering temperature on ITO/AgIn/ITO composite films. Such a study has not been reported to date.

In this work, ITO/AgIn/ITO composite films were prepared by magnetron sputtering. The current work focuses on the heat resistance of ITO/AgIn/ITO composite films upon changing the sputtering temperature in the preparation stage. The effect of sputtering temperature on their optical and electrical properties was systematically investigated. In addition, how the microstructure and surface morphology affect the optical and electrical properties are explained by characterization. The results provide proof for studying the thermal stability of silver alloy composite films, so as to better understand its influence.

## 2. Materials and Methods

High-purity ITO ceramic targets (99.99% purity, In_2_O_3_:SnO_2_ = 90 wt.%:10 wt.%) were used for the top and bottom ITO films, while for step ii, a target consisting of silver–indium alloy (Ag: 99 wt.%; In: 1 wt.%) was used, which undergoes melting, forging, rolling, heat treatment and machining processes to finally obtain finished targets of 50 mm diameter. The base pressure was 5 × 10^−4^ Pa and the working pressure was maintained at 0.7 Pa. ITO layers were prepared by RF sputtering at 50 W and the DC power for the middle AgIn alloy layer deposition was kept at 20 W. Prior to sputtering, both the ITO target and AgIn alloy target were pre-sputtered for 5–10 min to remove contaminants from the surface. The entire sputtering process was performed with high-purity argon gas. To investigate the effect of sputtering temperature on ITO/AgIn/ITO multilayer films, the multilayer films were deposited without breaking the vacuum, and six samples with different sputtering temperatures were obtained, employing temperatures of 25 °C, 100 °C, 150 °C, 200 °C, 250 °C and 300 °C.

A series of ITO/AgIn alloy/ITO multilayer films were prepared on glass substrates by magnetron sputtering at different sputtering temperatures, following a three-step procedure: (i) a 10 nm thick ITO film was deposited on a glass substrate, (ii) a 100 nm thick AgIn alloy layer was deposited on a glass substrate covered with ITO and finally (iii) an ITO top layer film was deposited for covering. The preparation principle and processing scheme of the ITO/AgIn/ITO multilayer films are shown in Figure 1.

The thickness of the ITO/AgIn/ITO multilayer films was measured with a step profiler. The crystal structure was determined using an X-ray diffractometer with Cu Kα radiation (λ = 1.5412 Å). The surface microstructure and cross-sectional observations of the multilayer films were characterized by field emission scanning electron microscopy (SEM). Surface roughness and surface morphology were estimated by atomic force microscopy (AFM). The electrical properties of the samples were measured using the van der Pauw method in a Hall effect measurement system. The sample was a square of approximately 10 × 10 mm^2^. For testing, the four contacts of the sample stage were pressed on the four corners of the sample at a distance of about 8.5 mm and then put into the tester for measurement. The contact distance could be adjusted according to the size of the sample. The applied test current was set to 10 μA. The test results were averaged over several measurements. Optical properties were evaluated with a UV-Vis spectrophotometer over the wavelength range of 800 to 300 nm.

## 3. Results

Figure 2 presents the XRD patterns of the ITO/AgIn/ITO films deposited on glass substrates at different sputtering temperatures. As shown in Figure 2, all samples had a broad peak with a two-theta angle from 20° to 30°, which could be attributed to the amorphous structure of the glass substrate. As for the ITO/AgIn/ITO films prepared with a sputtering temperature below 200 °C, specific diffraction peaks at 38.16°, 44.19°, 64.44° and 77.48° can be clearly observed, which correspond to the (111), (200), (220) and (311) planes of Ag, respectively. The XRD patterns can be well indexed with PDF #87-0717. When the sputtering temperature is increased to 300 °C, there is an additional diffraction peak at 2θ of 81.73°, which is indexed to be the (222) plane of the Ag. No diffraction peaks for In can be observed for all the ITO/AgIn/ITO films, suggesting that the In is fully dissolved in the Ag layer. This is due to the fact that, according to the Ag–In binary phase diagram, the theoretical solid solubility of In in the silver lattice is high (21 wt.%) [13]. AgIn with the addition of 1.0 wt.% In should be a solid solution phase, theoretically. As shown in Figure 3, no diffraction peaks of In were present in the XRD patterns of the samples at each stage of AgIn alloy target processing.

With the increase of sputtering temperature, the intensity of each diffraction peak of the ITO/AgIn/ITO films was significantly enhanced, and the films were oriented with [111] as the preferred orientation. This is due to the increase of the substrate temperature reaching the growth kinetic energy of the films, so that the deposited atoms have a certain diffusion ability, and the formed film layer is preferentially oriented along the [111] direction. At sputtering temperatures of 250–300 °C, diffraction peaks appeared at 2θ = 30.59° and 35.33°, which are attributed to the (222) and (400) diffraction peaks of In_2_O_3_ (PDF #06-0416). Similar results were observed by Yalan Hu et al., who prepared ITO films at different temperatures [26]. It was shown that as the temperature increased, the ITO films changed from the amorphous to the crystalline state, which improved the films in terms of structural defects. On the other hand, a comparison between AgIn/Glass (Appendix A), ITO/Glass (Appendix A) and the ITO/AgIn/ITO composite films (Figure 2 and Appendix A) in this work show that the intermediate layer of the AgIn film promotes the nucleation and crystallization of the top ITO film.

Figure 4 depicts the reflectance spectra of the ITO/AgIn/ITO films as a function of sputtering temperature. At a certain sputtering temperature, the reflectance gradually increases with increasing wavelength. The reflectance decreases sequentially with increasing sputtering temperature at 550 nm. The films prepared at 25 °C had the highest reflectance, up to 95.3% at 550 nm and 98.6% at 780 nm. The films prepared at 100 °C had an approximate reflectance, but the average reflectance in the visible wavelength range of 380–780 nm was slightly lower than the reflectance at 25 °C. The reflectance at 550 nm at 150 °C, 200 °C, 250 °C and 300 °C was 92%, 78.8%, 68.6% and 69.3%, respectively.

Figure 5 compares the resistivity, mobility and carrier concentration of the ITO/AgIn/ITO films at different sputtering temperatures. ITO/AgIn/ITO films had a low resistivity of about 6.7–7.8 μΩ·cm within the temperature range of 25–300 °C. The sheet resistance varied with the sputtering temperature in a small range of 0.58–0.66 Ω/sq. This is because the top ITO layer, the middle Ag layer and the bottom ITO layer in the ITO/AgIn/ITO films can be regarded as parallel structures, with the following relationship [27,28]:(1)$$\frac{1}{R_{ITO/AgIn/ITO}}=\frac{1}{R_{ITO(Top)}}+\frac{1}{R_{AgIn}}+\frac{1}{R_{ITO(Bottom)}},$$
where *R_ITO/AgIn/ITO_*, *R_ITO(Top)_*, *R_AgIn_* and *R_ITO(Bottom)_* are the square resistance of the ITO/AgIn/ITO composite film, the top ITO film, the AgIn film and the bottom ITO film, respectively, in units of Ω/sq. The resistance of the composite films depends mainly on the resistance of the middle Ag layer as can be seen from Equation (1). The resistivity of ITO/AgIn/ITO films showed an overall increasing trend with the increase of sputtering temperature. However, the resistivity of the films decreased slightly at 200 °C. This may be due to the fact that the grain size starts to increase significantly at 200 °C, the crystallization of the films increases, the carrier density increases and the concentration of ionized impurities follows, leading to a decrease in mobility. The relationship between carrier concentration, mobility and resistivity is given by Equation (2).
(2)$$\rho =\frac{1}{ne\mu },$$
where *ρ* is the resistivity, *n* is the carrier concentration, *μ* is the mobility and *e* is the electron charge. Since the carrier concentration increases faster than the mobility decreases, the resistivity decreases. The resistivity continued to increase again at sputtering temperatures of 250 °C and 300 °C, probably due to changes in the structure of the silver caused by the high sputtering temperature.

The reflectance and resistivity variations are mainly determined by the grain size and surface morphology of the films. Figure 6 shows the SEM images of ITO/AgIn/ITO films prepared at different sputtering temperatures. A significant change in the morphology of the films can be seen. At a sputtering temperature below 100 °C, the ITO/AgIn/ITO composite film had a flat and smooth surface with fine and uniform grains. The grain size of the composite film increased with the increase in sputtering temperature. The film agglomeration behavior was gradually enhanced. The grain size increased sharply at a sputtering temperature of 200 °C. Film island formation was noticeable and the holes were further increased at 300 °C.

The cross-sectional images of ITO/AgIn/ITO films are shown in Figure 7. With the increase of sputtering temperature, there is a significant change in the surface flatness state of the silver alloy films. The composite films prepared at a sputtering temperature below 100 °C had a flat continuous film surface. The film grains grew in a columnar structure. At a temperature of 200 ℃, the film still grew in a columnar structure at the beginning of sputtering. With the increase of sputtering time, the grains on the surface of the film agglomerate into hills. This is due to the fact that metal films deposited on substrates using sputtering methods are generally formed in an island growth mode, i.e., nucleation growth–island formation–continuous film formation [29,30]. The film thickness increased with the increase of sputtering time, the film grains agglomerated to form islands under the effect of temperature, the film surface flatness degraded and the surface roughness increased. The film thickness increases with the increase of sputtering time, the film grains agglomerate to form islands under the effect of temperature, the film surface flatness degrades and the surface roughness increases. At a sputtering temperature of 300 °C, Ag grains were affected by temperature, and agglomeration began to appear at the early stage of sputtering. The channels and voids between the islands increased, and the surface roughness of the films increased, which affected the optoelectronic properties of the films.

Film agglomeration is a phenomenon of nucleation and growth of holes; under certain thermodynamic conditions, continued agglomeration, i.e., growth and then impact of holes after formation, leads to the formation of islands [31,32]. A large number of Ag particles exist in the form of islands, and the channels and voids between the islands increase at higher temperatures. The continuity of the film layer is disrupted and the conductivity of the film layer decreases. Due to the scattering effect of Ag particles and holes, the light transmission between them is enhanced, which leads to a decrease in reflectivity.

To further investigate the effect of surface roughness on the reflectivity and resistivity of ITO/AgIn/ITO films, we checked their surface roughness and surface morphology. When the sputtering temperature was 25 °C, the films were dense and uniform, the surface was flat and smooth and the root means square (RMS) roughness of the surface was 3.38 nm. With the increase of sputtering temperature, the particle size increased, the surface roughness of the films increased and the continuity of the films was destroyed. Films prepared at 300 °C had high surface roughness and serious agglomeration phenomenon. The surface RMS roughness of the ITO/AgIn/ITO films increased from 3.38 nm at 25 °C to 40.26 nm at 300 °C (Figure 8), which was consistent with SEM image results. It indicates that with the increase of sputtering temperature, the flatness of the surface of the film degrades and the surface roughness increases, which directly leads to the enhancement of scattering, resulting in the decrease of reflectivity [33].

## 4. Discussion

To complete this work and clarify the performance advantages of the ITO/AgIn/ITO composite films, we compared their optical and electrical properties with those of related films reported in the literature [22,34,35,36], as shown in Table 1. In terms of optical properties, the ITO/AgIn/ITO composite film has the highest average reflectance compared to the AgIn film in the literature [22] in the visible range. The reflectance of the ITO/AgIn/ITO composite film deposited at 25 °C is higher than the related film reported in the literature, but slightly lower than the Ag film on the glass substrate in the literature [36]. The reflectivity of the composite films at a sputtering temperature of 250 °C is better than that of the Ag/Glass films. In terms of electrical properties, the resistivity of the ITO/AgIn/ITO composite film is slightly higher than that of the Ag/Glass film, but significantly better than that of ITO/Ag/ITO, ITO/Ag and AgIn films reported in the literature. This may be due to the addition of In, which has improved the heat resistance of ITO/AgIn/ITO, in the bottom ITO/Glass substrate it promotes the growth of AgIn and inhibits AgIn agglomeration [37] and it has improved the reflectivity and conductivity of ITO/AgIn/ITO composite film. In a nutshell, the ITO/AgIn/ITO composite film has excellent combined optical and electrical properties.

## 5. Conclusions

In summary, this work reports that sputtering temperature has a significant effect on the optical and electrical properties of ITO/AgIn/ITO composite films. The ITO/AgIn/ITO composite film deposited at a sputtering temperature of 25 °C exhibited the highest reflectance in the visible wavelength range, and composite films sputtered at 25 °C and 100 °C both had a reflectance of 95.3% at 550 nm and a resistivity of about 6.8–7.3 μΩ·cm. With the increase of sputtering temperature, the reflectivity gradually decreased and the resistivity slightly increased. There was a significant decrease in reflectivity at 200 °C, and the resistivity remained at 6.9 μΩ·cm. SEM images and AFM images demonstrated that the increase in temperature caused the surface grains of the films to agglomerate, the island structure to increase, the flatness of the film surface to degrade and the RMS roughness to increase. The properties of the composite films, especially the electrical and optical properties, depend mainly on the metal layers. This further illustrates that the reflectivity and resistivity of the films are closely related to the microstructure and surface morphology. It is shown that the ITO/AgIn/ITO thin films have a promising application for high-reflectivity anodes.

## Figures and Tables

**Figure 1 materials-16-02849-f001:**
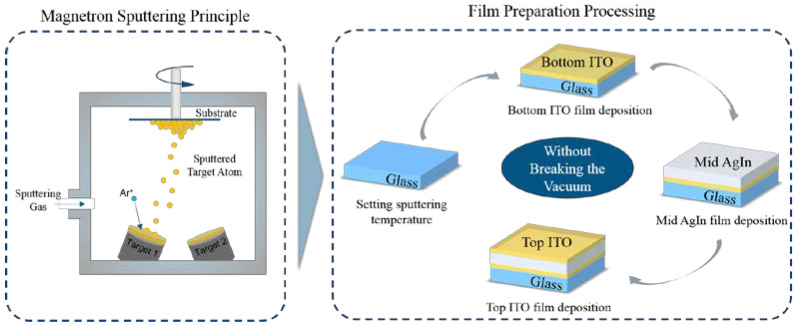
Scheme of ITO/AgIn/ITO multilayer film preparation principle and processing.

**Figure 2 materials-16-02849-f002:**
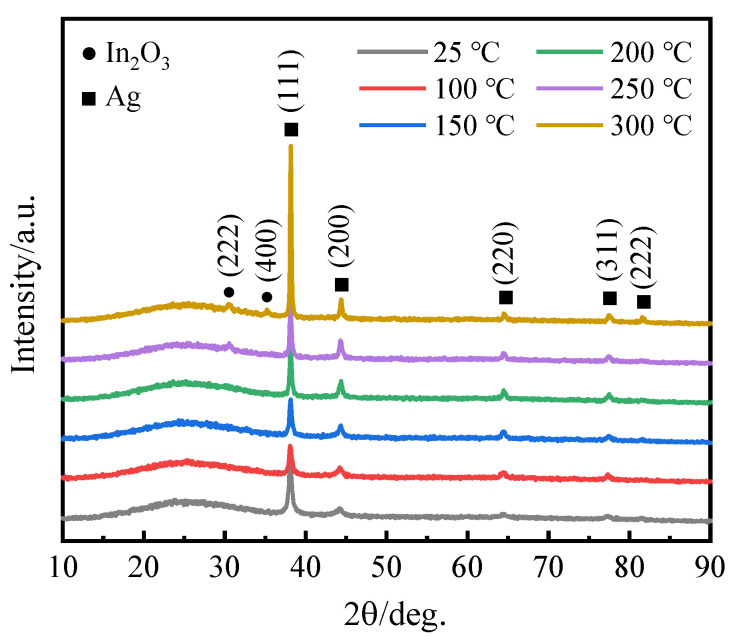
XRD patterns of the ITO/AgIn/ITO films at different sputtering temperatures.

**Figure 3 materials-16-02849-f003:**
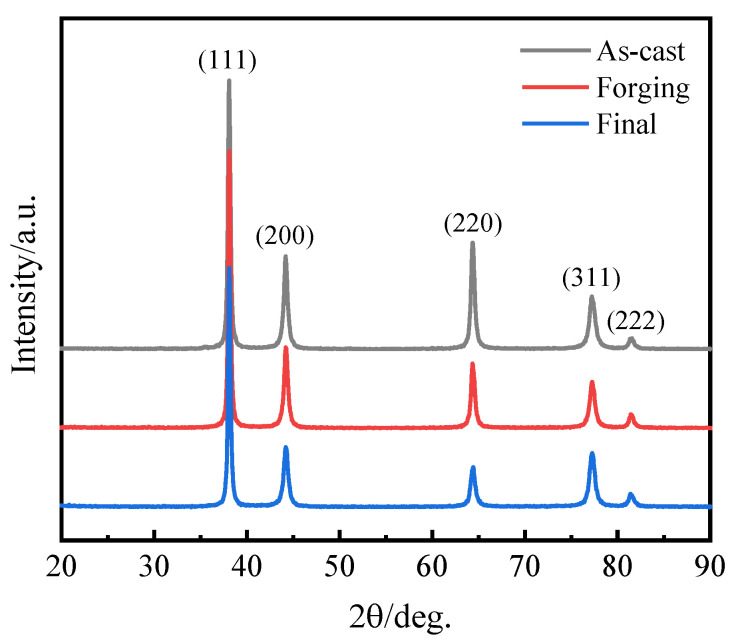
XRD patterns of as-cast, forging and final AgIn alloy targets.

**Figure 4 materials-16-02849-f004:**
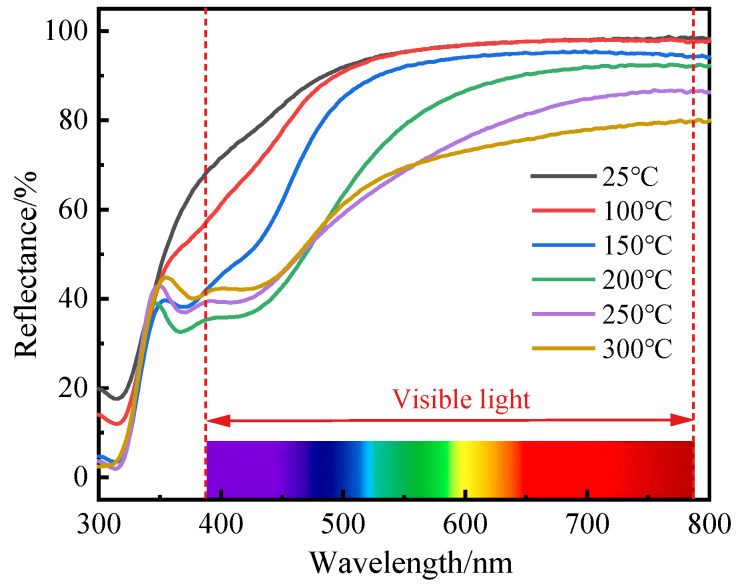
Reflectance spectra of the ITO/AgIn/ITO films as a function of sputtering temperature.

**Figure 5 materials-16-02849-f005:**
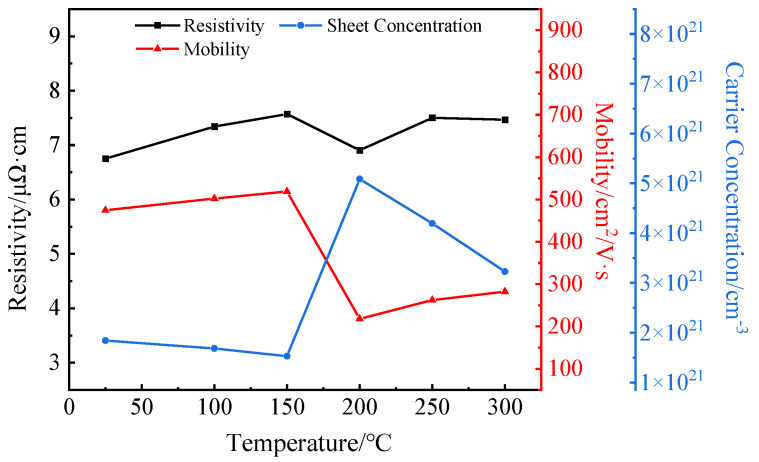
Hall measurement of the ITO/AgIn/ITO films as a function of sputtering temperature.

**Figure 6 materials-16-02849-f006:**
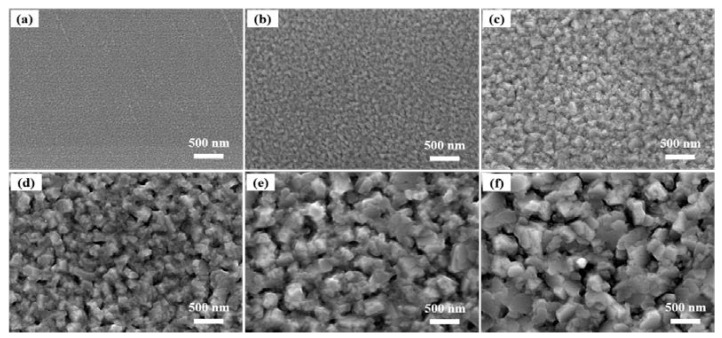
SEM images of the ITO/AgIn/ITO films at different sputtering temperatures. (**a**) 25 °C, (**b**) 100 °C, (**c**) 150 °C, (**d**) 200 °C, (**e**) 250 °C and (**f**) 300 °C, respectively.

**Figure 7 materials-16-02849-f007:**
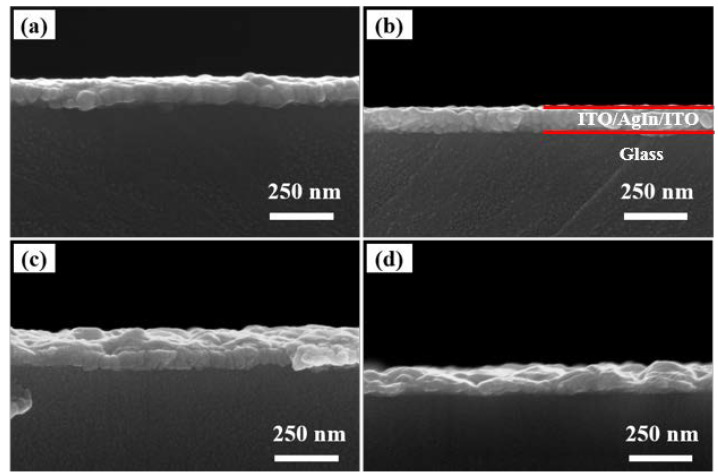
Cross-sectional SEM images of the ITO/AgIn/ITO films at different sputtering temperatures. (**a**) 25 °C, (**b**) 100 °C, (**c**) 200 °C and (**d**) 300 °C, respectively.

**Figure 8 materials-16-02849-f008:**
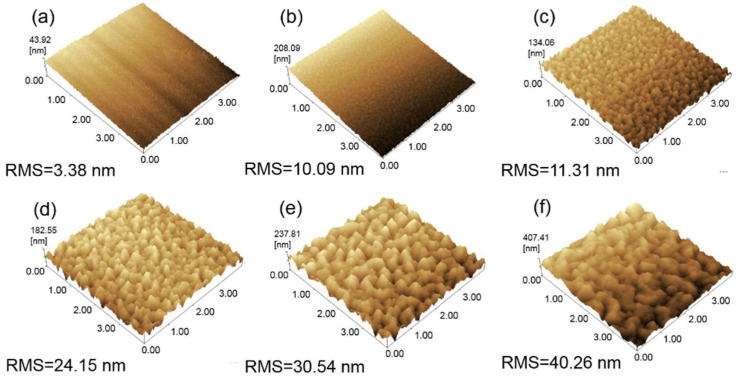
AFM images of the ITO/AgIn/ITO films at different sputtering temperatures. (**a**) 25 °C, (**b**) 100 °C, (**c**) 150 °C, (**d**) 200 °C, (**e**) 250 °C and (**f**) 300 °C, respectively.

**Table 1 materials-16-02849-t001:** Comparison of reflectance and resistivity of ITO/AgIn/ITO composite films with related films reported in the literature.

	Temperature (°C)	Reflectance at 550 nm (%)	Resistivity (Ω·cm)	References
ITO/AgIn/ITO	25	95.3	6.8 × 10^−6^	Present work
		91.8 ^1^		
	250	68.6	7.4 × 10^−6^	
AgIn reflector	25	84.0 ^1^	1.9 × 10^−5^	[12]
ITO/Ag/ITO	25	-	3.5 × 10^−5^	[34]
ITO/Ag	25	92.5	5.3 × 10^−5^	[35]
Ag/Glass	25	97.5 ^2^	3.7 × 10^−6^	[36]
	250	5 ^2^	5.0 × 10^−6^	

^1^ Average reflectance in the visible range. ^2^ Reflectance at 550 nm in the visible range.

## Data Availability

The data presented in this study are available on request from the corresponding author. The data are not publicly available due to the privacy of this research.

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
