# Peer review of "Influence of the Sputtering Temperature on Reflectivity and Electrical Properties of ITO/AgIn/ITO Composite Films for High-Reflectivity Anodes"

_materials, 2023, doi:10.3390/ma16072849_

Round 1
Reviewer 1 Report
The manuscript concerns ITO/AgIn/ITO composite films deposited by magnetron sputtering at different temperatures. The authors obtained a series of six samples that were examined structurally, optically and electrically. The obtained results indicate an increase of the RMS roughness with increasing temperature caused by progressive agglomeration of the thin layer. This in turn deteriorates the optical and electrical properties of the system. I agree with the authors that the influence of sputtering temperature on the particular ITO/AgIn/ITO composite films has not been reported yet in other papers, however all the effects described in the paper are expected and they don't show any new behaviour or novel phenomena.
The manuscript is clearly written and the research is solid. The results are presented in a well-structured manner. In my opinion, the work meets the standards of the journal and I have only a few minor comments listed below.
11. Please specify the size of the samples and the distances between the contacts in the van der Pauw method. Please also describe the method by which the contacts were made. Were they pressed on or permanently attached to the sample?
2. Page 3, line 94. Please replace "van der Paul method" with "van der Pauw method"
3. Page 3, line 106. The authors claim that “No diffraction peaks for In can be observed for all the ITO/AgIn/ITO films, suggesting that the In is fully dissolved in the Ag layer.” Due to the low content of indium and the small thickness of the layer, the In peak may be invisible due to the weak signal. Please discuss such possibility in the paper.
4. Please discuss if the 10 nm thick ITO top layer is continuous in the case of the samples deposited at the highest temperatures. For these samples, the RMS roughness can be expected to be greater than ITO layer thickness, so the continuity of the layer cannot be guaranteed. Then, there is probably no longer ITO/AgIn/ITO multilayer, but a mixed system. Please discuss this issue in the text of the paper.
5. Please enlarge the scale in Figure 7 to make visible the size of the measured areas.
Author Response
Comments 1: Reviewer #1: The manuscript concerns ITO/AgIn/ITO composite films deposited by magnetron sputtering at different temperatures. The authors obtained a series of six samples that were examined structurally, optically and electrically. The obtained results indicate an increase of the RMS roughness with increasing temperature caused by progressive agglomeration of the thin layer. This in turn deteriorates the optical and electrical properties of the system. I agree with the authors that the influence of sputtering temperature on the particular ITO/AgIn/ITO composite films has not been reported yet in other papers, however all the effects described in the paper are expected and they don't show any new behaviour or novel phenomena. The manuscript is clearly written and the research is solid. The results are presented in a well-structured manner. In my opinion, the work meets the standards of the journal and I have only a few minor comments listed below.
Response: Thanks a lot for the referee’s positive comment. We are sorry for not presenting the novelty of the present work. Silver films are bottom high-reflectivity anodes in the organic light-emitting devices, during the actual production and service the Ag films will undergo a high-temperature atmosphere. The agglomeration behavior of Pure-Ag films under high temperatures results in a decrease of reflectivity. It was expected that the addition of In elements into the Ag films effectively suppresses the agglomeration behavior of Ag films. However, the effect of sputtering temperature on the performance of ITO/AgIn/ITO composite films during the preparation of the samples has not been reported yet. Therefore, the present work focuses on the influences of sputtering temperature on the performance of ITO/AgIn/ITO composite films. Although the results are expected, it offers direct evidence for studying the heat resistance of silver alloy composite films. A better understanding of the temperature dependence of the optical and electrical properties, structure and morphology of ITO/AgIn/ITO composite films can be obtained. We have revised the motivation of the present work in the revised manuscript.
Comments 2: Please specify the size of the samples and the distances between the contacts in the van der Pauw method. Please also describe the method by which the contacts were made. Were they pressed on or permanently attached to the sample?
Response: Thanks a lot for the referee’s comment. The electrical properties of the samples in this study were measured by the van der Pauw method. The sample is a square of approximately 10 × 10 mm2. For the measurement of electrical properties, the four contacts of the sample stage are pressed on the four corners of the sample at a distance of about 8.5 mm and then put into the tester for measurement. The contact distance can be adjusted according to the size of the sample. The applied test current was set to 10 μA. The test results were averaged over several measurements. We have specified the size of the sample and the contact distance in the van der Pauw method in the revised manuscript.
Comments 3: Page 3, line 94. Please replace "van der Paul method" with "van der Pauw method".
Response: In the light of the referee’s comment, we have corrected the typos in the revised manuscript.
Comments 4: Page 3, line 106. The authors claim that “No diffraction peaks for In can be observed for all the ITO/AgIn/ITO films, suggesting that the In is fully dissolved in the Ag layer.” Due to the low content of indium and the small thickness of the layer, the In peak may be invisible due to the weak signal. Please discuss such possibility in the paper.
Response: Thank you very much for your suggestion. We agree with the reviewer’s comment that the low content of indium and the small thickness of the layer may result in the absence of diffraction peaks of In elements. Actually, the solid solubility of In element in the silver lattice is as high as 20 wt.%. In the present work, we added 1.0 wt.% In element into the Ag alloys. The AgIn alloy is of a solid solution phase, without any secondary phase of In. In order to check there is no secondary phase in the AgIn alloy, we also characterized the crystal structure and microstructure of the AgIn alloy target materials. As shown in Figure R1.1, only the diffraction peaks of Ag can be observed in XRD patterns of the AgIn targets. Therefore, we believe the added In elements are fully soluted in the AgIn alloys.
Comments 5: Please discuss if the 10 nm thick ITO top layer is continuous in the case of the samples deposited at the highest temperatures. For these samples, the RMS roughness can be expected to be greater than ITO layer thickness, so the continuity of the layer cannot be guaranteed. Then, there is probably no longer ITO/AgIn/ITO multilayer, but a mixed system. Please discuss this issue in the text of the paper.
Response: Thank you very much for the valuable reminder. We agree with the possibility you put forward. According to Figure R1.2, we can see that the ITO/Glass film prepared at 300 ℃ is considered to be continuous because of its low resistivity. But the continuity of the top ITO in the composite film cannot be determined now. Because the ITO film is too thin, the current scanning electron microscope images cannot be determined to be continuous. We will conduct follow-up research through transmission electron microscopy and other characterization techniques in future work.
Comments 6: Please enlarge the scale in Figure 7 to make visible the size of the measured areas.
Response: According to the referee’s suggestion, we have enlarged the scale of AFM images, as shown in Figure R1.3 and Figure 8 in the revised manuscript.

Reviewer 2 Report
This paper is devoted to the preparation and systematical investigation of the composite ITO/AgIn/ITO films. The preparation of the films was performed by magnetron sputtering, and the influences of sputtering temperature (ranged from 25 °C to 300 °C) on the optical and electrical properties of the films was the central point of this work.
The methods which were used for the material's characterization are adequate to the task and the general consideration of the manuscript show that it is can be considered for the publication in the Materials journal, however several minor points should be addressed first.
1) Throughout the discussion authors describe the influence of the sputtering temperature on the morphological characteristics of the films however, on the other side, the prepared films are composed of two new layers namely AgIn and ITO. It can be assumed that authors describe the cooperative effect on both layers. In this context, it is not clear how the action of different sputtering temperature influences each layer on its own? And also, how the morphology of AgIn layer influences the properties of newly formed ITO layer?
2) The properties of the prepared composite materials should be compared with the corresponding properties (XRD, Hall measurements and optical properties) of the parent ITO/glass substrates.
3) In the cross-sectional SEM images each layer should be appropriately assigned or marked.
Author Response
Comments 7: Reviewer #2: This paper is devoted to the preparation and systematical investigation of the composite ITO/AgIn/ITO films. The preparation of the films was performed by magnetron sputtering, and the influences of sputtering temperature (ranged from 25 °C to 300 °C) on the optical and electrical properties of the films was the central point of this work.
The methods which were used for the material's characterization are adequate to the task and the general consideration of the manuscript show that it is can be considered for the publication in the Materials journal, however several minor points should be addressed first.
Response: Thanks a lot for the referee’s comment, we have revised the minor points in the revised manuscript in terms of the referee’s comment.
Comments 8: Throughout the discussion authors describe the influence of the sputtering temperature on the morphological characteristics of the films however, on the other side, the prepared films are composed of two new layers namely AgIn and ITO. It can be assumed that authors describe the cooperative effect on both layers. In this context, it is not clear how the action of different sputtering temperature influences each layer on its own? And also, how the morphology of AgIn layer influences the properties of newly formed ITO layer?
Response: Thank you very much for your valuable comments. Indeed, we have investigated the single layer of AgIn and ITO films deposited on the glass substrate. Figure R2.1 shows the effect of sputtering temperature on XRD, optical and electrical properties of AgIn/Glass films. It is seen that the reflectance of the AgIn/Glass film decreases with increasing sputtering temperature and the resistivity generally shows an increasing trend.
Figure R2.2 shows the effect of sputtering temperature on XRD, optical and electrical properties of ITO/Glass films. The ITO/Glass film shows the opposite properties to the AgIn/Glass film, i.e. the transmittance gradually increases and the resistivity gradually decreases as the sputtering temperature increases.
As for the influence of AgIn layer on the top ITO layer, it can be seen from Figure R2.3 that ITO/Glass prepared at 200-300℃ was amorphous, while ITO/AgIn/ITO films at corresponding temperatures have (222) (400) diffraction peaks, indicating that the intermediate layer of AgIn films promotes the nucleation and crystallization of the top ITO films.
Comments 9: The properties of the prepared composite materials should be compared with the corresponding properties (XRD, Hall measurements and optical properties) of the parent ITO/glass substrates.
Response: Thank a lot the referee’s comment. We have compared the properties of the composites (Figures 2, 4 and 5 in this work) with the relevant XRD patterns, electrical and optical properties of the ITO/glass films (Figure R2.2). It is worth noting that the intermediate AgIn layer plays an important role in the structure and properties of the composite films. Figure R2.4 shows the comparison of SEM images of ITO/AgIn/ITO films and AgIn films at different sputtering temperatures. It can be clearly seen that the bottom ITO film has an inhibitory effect on the aggregation of the film, promotes the growth of Ag alloy film and makes the film layer more uniform.
Comments 10: In the cross-sectional SEM images each layer should be appropriately assigned or marked.
Response: Thank you very much for your suggestion. We have marked the cross-sectional images, as shown in Figure R2.5 and Figure 7 in the revised manuscript. However, due to the thickness of ITO film is only 10nm and the large RMS roughness of AgIn layer, the structure and morphology of each layer of the composite film cannot be distinguished by SEM. In the future work, we will use a transmission electron microscope and other characterization methods to study.

Reviewer 3 Report
In this work, ITO/AgIn/ITO composite films were prepared by magnetron sputtering, and the effect of sputtering temperature on their optical and electrical properties was systematically investigated. In addition, how the microstructure and surface morphology affect the optical and electrical properties are explained by characterization. The paper is interesting but its motivation and authors' contribution are not clear.
What is the main task of such investigation? This point is not clear.
Moreover, the paper needs part 'Discussion' to discuss the results from all points of view and to present further research directions.
The authors should present a scheme or workflow of their experiment that was mentioned in lines 84-88.
Table 1 should be better commented on.
Author Response
Comments 11: Reviewer #3: In this work, ITO/AgIn/ITO composite films were prepared by magnetron sputtering, and the effect of sputtering temperature on their optical and electrical properties was systematically investigated. In addition, how the microstructure and surface morphology affect the optical and electrical properties are explained by characterization. The paper is interesting but its motivation and authors' contribution are not clear. What is the main task of such investigation? This point is not clear.
Response: Thank you for your comments. We have added the main task of the present work in the revised manuscript. The film will be baked and heated in actual production. Ag films are subject to agglomeration behavior when sputtered at higher temperatures. AgIn alloy with the addition of In element can effectively inhibit silver agglomeration. Therefore, the current work focuses on the heat resistance of ITO/AgIn/ITO composite films directly by changing the sputtering temperature in the preparation stage. Through the analysis of optical and electrical properties, structure and morphology, provides proof for studying the thermal stability of silver alloy composite films, so as to better understand its influence.
Comments 12: The paper needs part 'Discussion' to discuss the results from all points of view and to present further research directions.
Response: Thanks for your suggestion. We have added new discussions in the relevant parts of the revised manuscript.
Comments 13: The authors should present a scheme or workflow of their experiment that was mentioned in lines 84-88.
Response: Thank you very much for your valuable comments. We have added a scheme of film preparation principles and processing, as shown in Figure R3.1 and Figure 1 in the revised manuscript.
Comments 14: Table 1 should be better commented on.
Response: Thank you for your advice. We have added the discussion in the revised manuscript.

Round 2
Reviewer 3 Report
The authors addressed all my concerns. The paper can be accepted.